# An ensemble deep learning framework for energy demand forecasting using genetic algorithm-based feature selection

Mohd Sakib[1], Tamanna Siddiqui [1,2]*, Suhel Mustajab[1], Reemiah Muneer Alotaibi [2], Nouf Mohammad Alshareef[2], Mohammad Zunnun Khan[3]

**1** Department of Computer Science, Aligarh Muslim University, Aligarh, UP, India, **2** Department of Information Technology, Imam Mohammad Ibn Saud Islamic University, Riyadh, KSA, **3** Department of Information Systems and Cybersecurity, University of Bisha, Bisha, KSA

* TZSiddiqui@imamu.edu.sa, tsiddiqui.cs@amu.ac.in

**Data Availability Statement:** Information is available in a manuscript file.

**Funding:** The author(s) received no specific funding for this work.

## Abstract

Accurate energy demand forecasting is critical for efficient energy management and planning. Recent advancements in computing power and the availability of large datasets have fueled the development of machine learning models. However, selecting the most appropriate features to enhance prediction accuracy and robustness remains a key challenge. This study proposes an ensemble approach that integrates a genetic algorithm with multiple forecasting models to optimize feature selection. The genetic algorithm identifies the optimal subset of features from a dataset that includes historical energy consumption, weather variables, and temporal characteristics. These selected features are then used to train three base learners: Long Short-Term Memory (LSTM), Bi-directional Long Short-Term Memory (BiLSTM), and Gated Recurrent Unit (GRU). The predictions from these models are combined using a stacking ensemble technique to generate the final forecast. To enhance model evaluation, we divided the dataset into weekday and weekend subsets, allowing for a more detailed analysis of energy consumption patterns. To ensure the reliability of our findings, we conducted ten simulations and applied the Wilcoxon Signed Rank Test to the results. The proposed model demonstrated exceptional precision, achieving a Root Mean Square Error (RMSE) of 130.6, a Mean Absolute Percentage Error (MAPE) of 0.38%, and a Mean Absolute Error (MAE) of 99.41 for weekday data. The model also maintained high accuracy for weekend predictions, with an RMSE of 137.41, a MAPE of 0.42%, and an MAE of 105.67. This research provides valuable insights for energy analysts and contributes to developing more sophisticated demand forecasting methods.

## 1. Introduction

In the modern era, accurate energy demand forecasting has become an essential component of efficient energy management. Energy consumption is rising for various reasons, such as population growth, increasing building energy needs, and expanding technological applications.

**Competing interests:** The authors have declared that no competing interests exist.

According to the data presented in [1], global energy consumption is expected to rise by approximately 70% by the year 2040. This alarming figure has necessitated the development of advanced predictive models to forecast energy demand, which are crucial for optimizing energy use in smart cities, thereby making them more efficient and sustainable. Machine learning models have become increasingly popular due to advancements in computational techniques and the availability of large amounts of data [2]. However, a critical challenge in these models is the effective selection of relevant features that enhance the accuracy and robustness of the predictions. To address this issue, we have implemented a Genetic Algorithm (GA) for optimal feature selection. The study first trains the model to determine the best features from the dataset with the help of GA; afterward, these features are fed into a stacking-based ensemble model for further training and evaluation.

At its core, energy demand forecasting may be broken down into two distinct ways: traditional methods and more modern machine learning-based techniques [3, 4]. Conventional methods include statistical analysis, Auto Regressive Integrated Moving Average (ARIMA), Exponential Smoothing (EA), and regression-based approaches [5]. These methods have proven effective for linear problems. On the other hand, machine learning methods are adept at handling non-linear scenarios. Notable among these are Random Forest (RF) [6], Decision Tree (DT) [7], and Support Vector Machines (SVM) [8], which have been utilized for energy demand forecasting, as discussed in recent studies. However, it is important to note that while machine learning models offer substantial advantages in time series prediction, they are also susceptible to limitations, such as the potential for becoming trapped in local minima, particularly if hyperparameters are not optimally tuned.

Several diverse methods have been developed for forecasting future energy consumption, from the very short term (minutes) to the very long term (years) (weeks) [9, 10]. Accurately predicting energy demand is crucial; an overestimation can lead to unnecessary conversion of excess energy, which is costly in terms of time, money, and resources, while underestimation may result in blackouts from supply line overloads. Typically, forecasting is divided into three categories based on the prediction horizon: short-term load forecasting (from an hour to a week), medium-term load forecasting (from a month to a year), and long-term load forecasting (beyond a year) [11]. Prediction of short-term loads is a major challenge. Indeed, schedules may be created to determine the distribution of generating resources, operational limits, environmental restrictions, and equipment usage restrictions with the help of accurate and dependable predictions [12]. In addition, the power systems may be optimized more using these predictions for the expected load condition in the future.

Key contributions of this paper are as follows:

- Developed a stacking-based ensemble deep learning model that combines the capabilities of Long Short Term Memory (LSTM), Bi-directional Long Short Term Memory (BiLSTM), and Gated Recurrent Unit (GRU) to improve the accuracy of forecasting.

- A genetic algorithm is employed to select optimal features, ensuring the model uses the most relevant feature for training and prediction.

- A detailed hyperparameter optimization method is used to find the best parameter values for the individual models, focusing on the "Epoch," which refers to the full recurrence over the training dataset.

- The dataset is stratified into distinct patterns for weekdays and weekends to facilitate a more nuanced analysis of energy usage trends. Further, for comprehensive and robust model validation, the dataset is divided into four subsets: S1, S2, S3, and S4

- To ensure the reliability of our findings, simulations were repeated ten times. We applied the Wilcoxon Signed Rank Test to these results, providing a statistically rigorous evaluation.

The evolution of the paper is organized as follows. After the introduction, the related work of the study has been explained in Section 2. Section 3 discusses the methodology used in this study. The experimental setup for the training of our model is then described in Section 4. Our proposed framework is presented in Section 5. The obtained results are carefully examined and discussed in Section 6. Finally, Section 7 presents a summary of the key findings, discusses their implications, and suggests directions for further investigation.

## 2. Related work

Electricity distribution networks are foundational to long-term prosperity and societal advancement. Power consumption forecasts must always be accurate and efficient when developing energy predictions for the dynamic electricity sector. In the dynamic electricity sector, power consumption forecasts must be both precise and efficient. Over recent decades, a myriad of methods for predicting future energy demand has emerged [13, 14]. Typically, these methods employ time series datasets of past energy usage to construct forecasting models [15]. The majority of the approaches presented for predicting a building's energy usage may be broken down into two classes: statistical and artificial intelligence-based. For forecasting and analyzing energy use in the future, statistical approaches use the available historical data to build probabilistic models. ARIMA, ARMA, and ARIMAX are well-known statistical methods that have been used to predict future energy consumption. However, AI-driven approaches can improve forecasting accuracy due to their ability to identify non-linear trends in time series data [16]. Despite the widespread use of feature selection techniques like correlation-based, filter, and wrapper methods in research, there remains a significant opportunity to incorporate more sophisticated techniques to refine these predictions further.

LSTM networks have demonstrated promising results in time series analysis [17]. Older variants of RNNs, such as Recurrent Backpropagation, require a very long time to learn and store the information over an extended period. To address these challenges, Hochreiter et al. [18] developed a gradient-based method called LSTM neural network [19], which is the extended version of a recurrent neural network. LSTM networks have both long-term memory and short-term memory, and it reduces the problem of exploding and vanishing gradient. Numerous researchers have leveraged the LSTM network to forecast energy demand. Bouktif et al. [20] integrated various machine learning models along with LSTM to develop a model for short-term load demand of energy.

In another study, the authors [21] presented a statistical model, SARIMA, to predict hourly wind speed in the coastal area of Scotland. They utilized three-time series datasets collected from different elevations. The accuracy of the forecasting model can be enhanced by using a combination of the homogeneous or heterogeneous models. Li et al. [22] considered this benefit and proposed a multi-energy forecasting method for energy systems using a fusion of a Convolutional Neural Network and GRU (CNN-GRU) with transfer learning on a parallel architecture. The performance of the model was further refined through hyperparameter tuning, leading to more accurate results. A summary of notable studies on demand forecasting is provided in Table 1.

Significant progress has been made in enhancing energy demand forecasting using deep learning models. However, a gap remains in optimizing feature selection within these models. To address this gap, we propose a two-tiered approach. The first step involves applying a genetic algorithm to identify and select the most influential features. In the subsequent step, we employ a stacking ensemble method that combines the strengths of multiple predictive

Table 1. Notable studies on demand forecasting using deep learning.

| Source | Year | Description/Findings | Techniques | Feature Selection | Limitations |
|---|---|---|---|---|---|
| Ullah et al. [23] | 2020 | This study explores various deep-learning methods for energy demand forecasting in smart cities. | DRL | No | Limited dataset availability for validation. |
| Xu et al. [24] | 2019 | This research presents a hybrid model combining time series analysis and deep learning. | LR, DBN | No | Computational complexity in training deep models. |
| Mohammad et al. [25] | 2019 | Authors investigate the use of RNNs for improving smart grid demand. | Deep-RNN | No | Limited consideration of external factors. |
| Ahmed et al. [26] | 2020 | The research presents an ensemble approach combining multiple forecasting models for energy demand prediction in smart cities. | ANN, EL | No | Complexity in model selection and tuning. |
| Taleb at al. [27] | 2022 | The performance of various hybrid deep learning models has been evaluated. | CNN, QRNN | Yes | Difficulty in model interpretability. |
| Choi et al. [28] | 2020 | The study demonstrates and reduces error rates for various facilities and highlights seasonal variations in residential power demand patterns. | LSTM | No | External factors were not taken into consideration |
| Ajayi et al. [29] | 2022 | In order to reduce energy inefficiency, this paper uses machine learning methods to predict annual energy use in residential structures during early design. | ANN, GB, DNN, KNN, DT, LR | No | The accuracy of these predictions could be enhanced by utilizing various factors. |
| [30] | 2020 | Employed Bi-directional LSTM, enhancing prediction accuracy compared to conventional LSTM models in mid-term forecasting. | BiLSTM | No | Requires large training datasets |
| Aslam et al. [31] | 2021 | The study emphasizes the importance of forecasting in managing the intermittent nature of RESs to ensure reliability. | DL, RES | No | The challenge of integrating vast datasets from diverse sources. |
| Kim et al. [32] | 2023 | This study presents a forecasting model to improve the operation of microgrids by addressing the uncertainties associated with DAP. | GA-AWPSO | Yes | Performance is highly dependent on the quality and granularity of the input datasets. |

DRL: Deep Reinforcement Learning; DBN: Deep Belief Network; EL: Ensemble Learning; QRNN: Quasi-Recurrent Neural Network; TL: Transfer Learning; GB: Gradient Boosting; DNN: Deep Neural Network; KNN: K-Nearest Neighbors; RES: Renewable Energy Sources; DAP: day-ahead prices.

models. This synergistic approach not only refines the feature set but also integrates various models to substantially improve the accuracy of energy demand forecasts.

## 2.1 Alternative approach

Numerous studies have explored data-driven techniques due to their ability to predict loads and energy consumption across various scenarios [33]. The standard data-driven approach typically minimizes the sum of squared vertical distances to determine the optimal parameters [34]. A widely used method is linear regression [35], which identifies the best-fitting straight line across the training data. In addition to the linear model, ridge regression penalizes the extreme values of the weighted matrix to deal with multi-collinearity [36]. There has been a noticeable improvement in the development and interpretation of these linear regression models. An example of a non-parametric model is the K-nearest neighbor, in which the forecast is just the mean of all the neighbors [37]. With the optimal parameter settings, this method provides accurate outcomes for electrical load forecasting; it is also relatively easy to understand and implement in practice. Rather than relying on a single decision tree, which can lead to overfitting, ensemble models employ a network of decision trees to make predictions, such as additional trees and random forests. Gradient boosting further enhances tree accuracy by employing an ensemble of weak learners that assign greater weight to misclassified predictions [38].

## 2.2 Problem formulation

The objective of this study is to develop an advanced method for predicting energy consumption. This is accomplished by integrating deep learning models using a stacking ensemble approach with a feature selection process optimized through a genetic algorithm (GA).

With GA, our goal is to find the optimal Chromosome $C^*$, that maximizes Fitness (C) as given in Eq (1)

$$C^* = \arg \max \textit{Fitness } (C) \qquad (1)$$

Where C = $[c_1, c_2, \ldots\ldots, c_n]$ is the chromosome, and $c_i$ is a binary variable indicating the inclusion (1) or exclusion (0) of feature $i$.

With stacking, our goal is to minimize the loss function $L(Y, \hat{Y}) = \frac{1}{N}\sum_{i=i}^{N}\left(Y_i - \hat{Y}_i\right)^2$ using the feature subset $X^{'}$ chosen by the GA and forecasts generated by the stacking model given in Eq (2)

$$\text{Minimize } L(Y, S(M_1(X'), (M_2(X'), \ldots\ldots, (M_m(X')))) \qquad (2)$$

$$\text{Subject to: } X' = \{x_i \mid c_i = 1, c_i \in C^*\}$$

Here $M_j(X^{'})$ represents the prediction output of the $j^{-th}$ model, and S is the stacking layer function that aggregates the prediction from the individual models.

In summary, the GA enhances the quality of the model input by optimizing the feature space, whereas the stacked model utilizes the advantages of numerous deep learning architectures to provide reliable prediction outputs.

## 2.3 Key challenges addressed by the proposed method

In this subsection, we outline the key challenges in energy demand forecasting and explain how our proposed method effectively addresses each challenge.

**2.3.1 Feature selection complexity.** The high-dimensional nature of datasets in energy demand forecasting presents a significant challenge in selecting the most relevant features [39]. This task is computationally intensive and often includes redundant or irrelevant features, which can degrade model performance. This study employs a GA to search for the optimal subset of features systematically. The GA uses a fitness function based on the inverse of Mean Squared Error to evaluate predictive accuracy given in Eq (3)

$$\textit{Fitness } (C) = \frac{1}{MSE(C)} \qquad (3)$$

Where C is a chromosome representing a subset of features.

**2.3.2 Model generalization.** A crucial challenge in energy demand forecasting is preventing overfitting by ensuring the model works effectively with unknown data. Overfitting occurs when the model captures noise in the training data, resulting in poor performance on new data [40]. By using the GA to determine the most informative features, our approach improves the model's generalizability.

**2.3.3 Computational efficiency.** The high-dimensional feature spaces might result in longer training times and more resource use due to the computational complexity. This challenge is significant in practical applications where computing efficiency is the highest priority. Our genetic algorithm-based approach addresses this difficulty by decreasing the number of features. The GA progressively improves the feature subset, reducing input space that retains the most significant features.

Let $n$ be the original number of features and k the number of selected features after GA optimization, where $k < n$.

## 3. Methods

In this section, we provide a concise overview of the methodologies employed in this study.

### 3.1 Time series analysis

This section presents an introduction to the basics of time series analysis. To get a more thorough analysis of time series analysis, we suggest reading [41, 42]. Time Series (TS) analysis is a valuable tool for studying energy demand patterns, which inherently vary over time due to their time-dependent nature. It involves applying statistical or artificial intelligence methods to predict future trends by analyzing previously collected information.

A time series composed of P real-valued data points $a_1, ...., a_P$ each $a_i$ (where $1 \leq i \leq$ P) signifies the value recorded at the time i. The task of time series forecasting can be described as predicting future values $a_{z+1}, .... a_{z+k}$ based on the preceding $a_1, ..., a_z$ values (where z + k $\leq$ P). The goal here is to reduce the difference between the forecasted value $\hat{a}_{z+j}$ and the actual value $a_{z+j}$ (for $1 \leq j \leq$ k). In this context, 'z' represents the historical window, indicating the number of past data points considered for making predictions, and 'k' denotes the prediction horizon, which is the extent of the future we aim to predict.

Time series analysis has been applied in a wide range of real-life applications such as Anomaly detection [43], financial indices [44], Healthcare [45], Weather prediction, and energy consumption [46]. In contrast to traditional approaches, which focus on parametric-based models, namely auto-regressive, Moving Average, and structural TS [47], the Artificial intelligence-based model offers purely data-driven approaches.

### 3.2 Genetic algorithm for feature selection

GA are adaptive heuristic search algorithms based on the evolutionary ideas of natural selection and genetics [48]. With this technique, our goal is to identify the subset of features that contributes the most to the predictive power of a given model. In this study, we employed the DEAP (Distributed Evolutionary Algorithms in Python) library, which provides a flexible and customizable framework for creating genetic algorithms. Below is a detailed description of the GA process for feature selection.

- **Chromosome Representation**

Let C = $[c_1, c_2, ......, c_n]$ be a chromosome, where n is the number of features and $c_i$ is a binary variable indicating the inclusion (1) or exclusion (0) of feature $i$.

$$C = [c_1, c_2, ...., c_n]$$

- **Fitness Function**

The fitness of an individual is evaluated based on the performance of the model using the selected subset of features. The Mean Squared Error (MSE) is used as the fitness metric

presented in Eq (4).

$$\text{Fitness (C)} = \frac{1}{N} \sum_{i=1}^{N} (y_i - \hat{y}_i)^2 \tag{4}$$

- **Genetic Operators**

  - **Selection:** Tournament selection is used to select individuals based on their fitness.

  - **Crossover:** Two-point crossover combines parts of two parent chromosomes to produce offspring.

  - **Mutation:** Flip-bit mutation changes bits in the chromosome with a certain probability.

- **Optimization Process**

  The goal is to find the optimal chromosome $C^*$ that maximizes Fitness (C)
  Formally:

  $$C^* = \arg \max_c Fitness(C)$$

  The steps employed for determining the best features are shown in Fig 1.

  The GA identifies the best subset of features from the dataset, which includes historical energy consumption, weather variables, and temporal features, by exploring various combinations to maximize predictive accuracy. The GA reduces redundancy and noise by choosing only the most relevant features, resulting in more accurate and robust model training. This optimized feature set enhances the model's performance, allowing it to generalize better from training data to unseen data, as shown by improved Root Mean Squared Error (RMSE), Mean Absolute Percentage Error (MAPE), and Mean Absolute Error (MAE) metrics. Furthermore, reducing the number of features decreases computational complexity, leading to faster training times without sacrificing accuracy.

## 3.3 Ensemble learning

Ensemble models have gained popularity in recent years mainly because of the great outcomes they produce on various tasks, such as classification and regression issues [49]. These techniques include the combination of multiple models of learning to enhance the performance of each model individually. Ensemble learning was first explored in the 1990s when it proved that many weak learners could be turned into strong learners.

Typically, two stages are involved in this process. Initially, a variety of base learners are developed from the training data. Subsequently, these learners are combined in the second stage to create a cohesive prediction model. This results in multiple forecasts derived from individual base learners being merged into a more effective composite model, which typically outperforms each of the base models. Consolidating several effective individual models into a single enhanced model usually results in greater predictive accuracy. Bagging, boosting, and stacking are the three most popular and well-known fundamental ensemble approaches [50].

In bagging, multiple models are created, with each model's results given equal weight, and a voting method determines the most common outcome. For regression, the mean of predictions is typically used. Boosting, similar to bagging, differs by assigning varying weights to models, with the final result being a weighted vote. In regression contexts, this means a weighted average. On the other hand, Stacking employs different algorithms for model

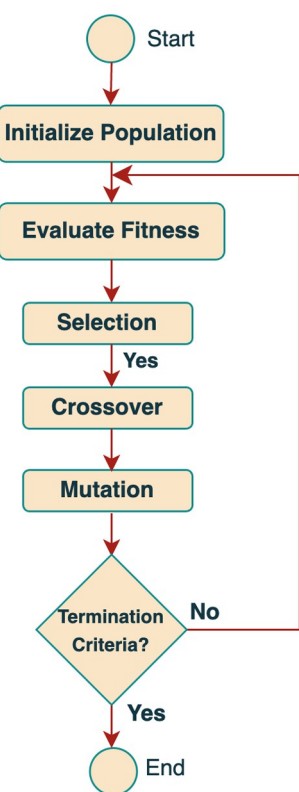

**Fig 1. Steps involved in genetic algorithm for feature selection.**

building, followed by a combiner algorithm that uses these models' outputs to make final predictions. Any ensemble approach can be used as the combiner in stacking.

Ensemble learning techniques have been applied in various time series forecasting methods, including energy demand. Zhang et al. [51] introduced an extreme learning machine (ELM) method utilized in the electricity market. Tan et al. [52] combined wavelet transform with ARIMA and GARCH models for price forecasting in Spanish and PJM electricity markets. In another study, authors [49] developed an ensemble model using Bayesian Clustering by Dynamics (BCD) and SVM, which was tested on New York City's historical load data. Later, Tasnim et al. [21] created an ensemble framework based on the cluster for predicting wind power, employing regression models on wind data from various Australian locations.

**3.3.1 LSTM.** LSTM is a specific form of RNN structure developed to tackle the challenge of the vanishing gradient problem and efficiently capture the patterns in sequential data [53]. This is achieved by utilizing a specialized gating mechanism that effectively controls the transmission of information within the network. A standard LSTM unit has three primary gates, as shown in Fig 2.

- Forget Gate ($F_t$): The forget gate is responsible for determining the retention or discarding of information from the previous cell state ($C_{t-1}$) Eq (5)

$$F_t = \phi\left(\mathrm{w}_t \times x_t + U_f \times h_{t-1} + b_f\right) \tag{5}$$

where w and b are weights and biases and $\phi$ represent activation function, which is sigmoid in our study. $U_f$ is a weight matrix associated with the forget gate.

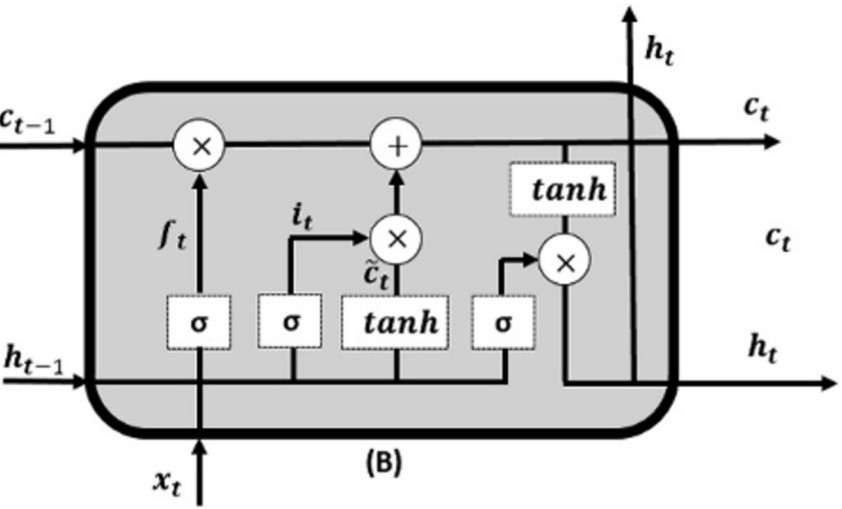

**Fig 2. The diagram of the LSTM neural network.**

- Input Gate ($i_t$): It decides what fresh information should be saved in the cell state. Eq (6)

$$i_t = \phi \left( w_i \times x_t + U_i \times h_{t-1} + b_i \right) \tag{6}$$

- Output Gate ($o_t$): Finally, the output gate manages what information should be passed to the next time step and what should be the final prediction. Eqs (7), (8), (9), and (10)

$$o_t = \phi \left( w_o \times x_t + U_o \times h_{t-1} + b_o \right) \tag{7}$$

$$c'_t = \tanh \left( w_c \times x_t + U_c \times h_{t-1} + b_c \right) \tag{8}$$

$$c_t = f_t \cdot c_{t-1} + i_t \cdot c'_t \tag{9}$$

$$h_t = o_t \cdot \sigma_c \left( c_t \right) \tag{10}$$

**3.3.2 BiLSTM.** Unlike standard LSTM, BiLSTM is able to learn from both the past and future as processes input in both directions. This feature enhances its capability to model sequential dependencies in language processing tasks. BiLSTM involves adding an additional LSTM layer that processes the input sequence in reverse. The outputs of both forward and backward layers are then merged using techniques like averaging, summing, multiplying, or concatenating. The unrolled BiLSTM structure is presented in Fig 3.

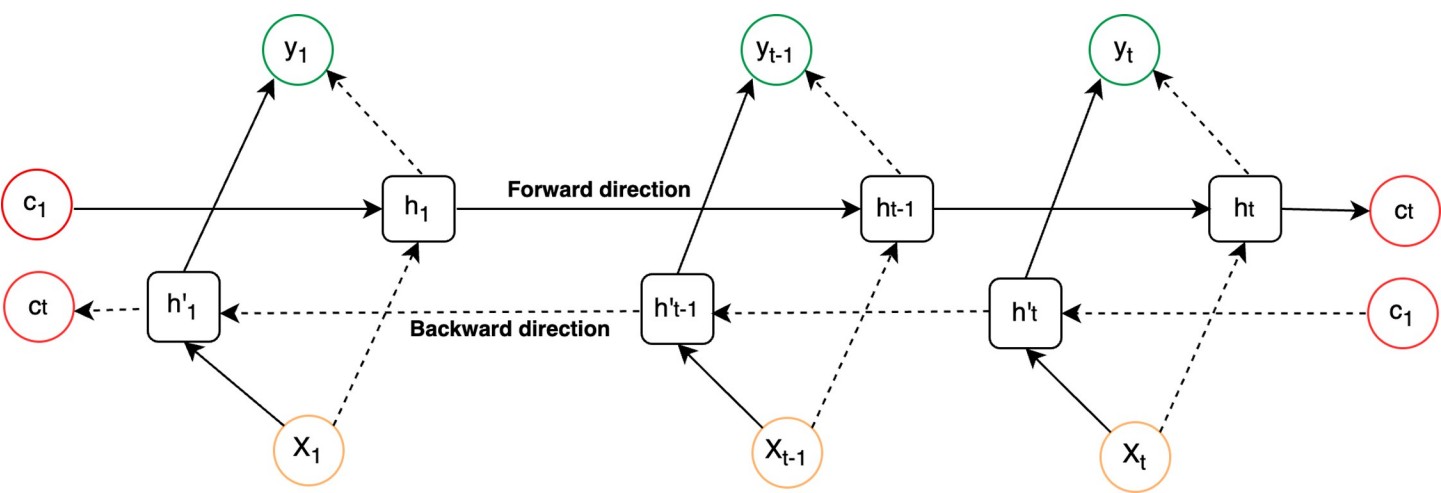

**Fig 3. A basic structure of the bidirectional long short-term memory.**

The forward and backward propagation output of the above diagram has been shown below: Eqs (11), (12), and (13).

$$\vec{h} = \overrightarrow{LSTM}\ (h_{t-1}, x_t, c_{t-1}) \tag{11}$$

$$\overleftarrow{h} = \overleftarrow{LSTM}\ (h_{t+1}, x_t, c_{t+1}) \tag{12}$$

$$H_t = \left[\vec{h}_t, \overleftarrow{h}_t\right] \tag{13}$$

Where x, h, and c are the input state, hidden state, and temporary state.

**3.3.3 GRU.** The GRU model demonstrates superior computational efficiency by employing a lower number of training parameters. As a result, memory utilization and training times are reduced. The simpler architecture, consisting of two gates, Eqs (14) and (15), reduces the possibility of overfitting on smaller datasets. Moreover, the GRU shows improved stability during the training process, requiring reduced fine-tuning for hyperparameters. Fig 4 shows a simple GRU network.

Update Gate ($Z_t$):

$$Z_t = \sigma(W_z \times x_t + U_z \times h_{t-1}) \tag{14}$$

Reset Gate ($R_t$):

$$R_t = \sigma(W_r \times x_t + U_r \times h_{t-1}) \tag{15}$$

where W are the weights, and σ represents the activation function. $h_{t-1}$ are the hidden states.

## 3.4 Stochastic model validation and statistical analysis

The stochastic nature of deep learning models such as LSTM, BiLSTM, GRU, and GA, where random initializations and training processes can lead to variability in performance, makes it crucial to ensure the reliability and robustness of our results [54]. We conducted repeated simulations to address this, running each model 10 times with different random seeds. We

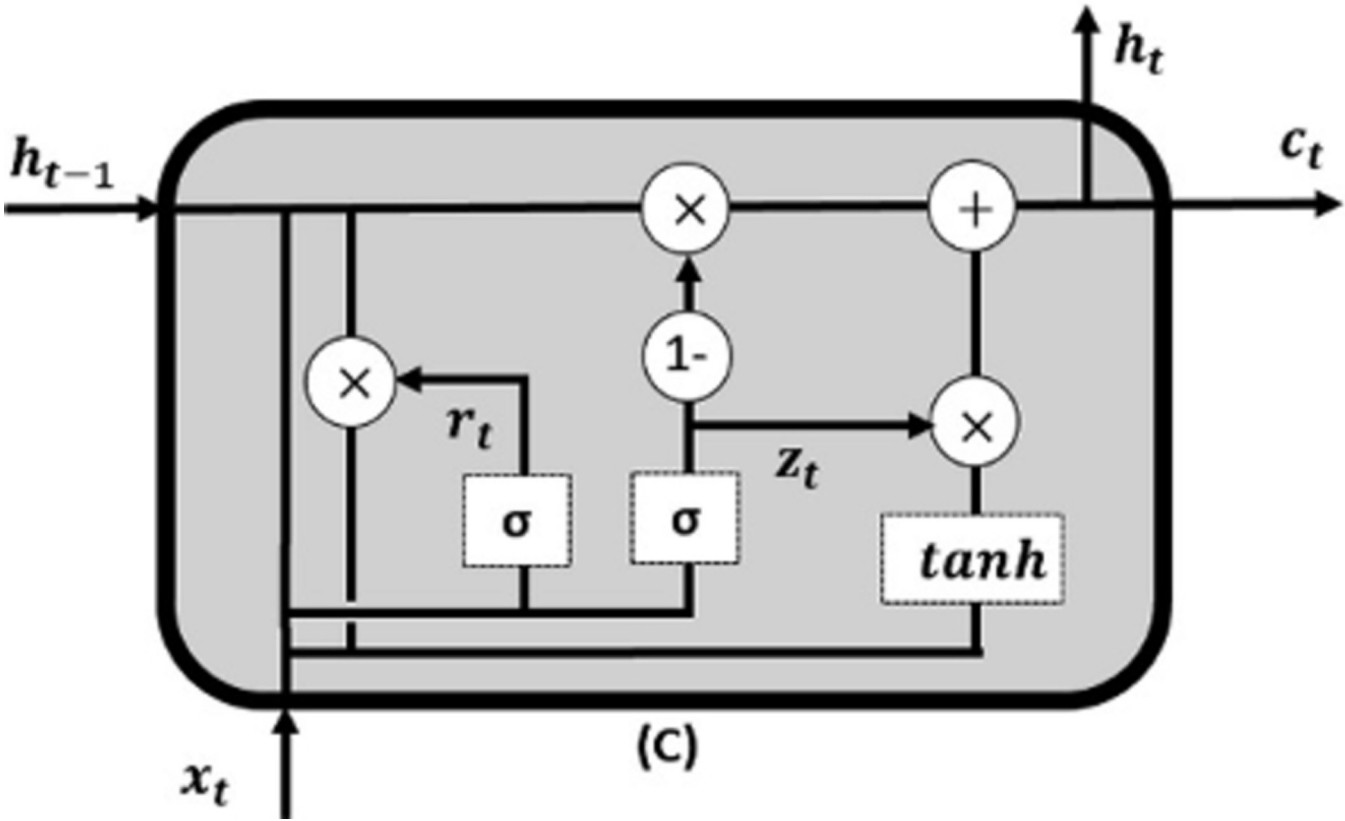

**Fig 4. A simple structure of GRU.**

recorded the performance metrics for each run, including MSE, MAE, MAPE, and RMSE. Eqs (18)–(21) provide the equations for these metrics.

To statistically compare the performance metrics across different runs, we employed the Wilcoxon Signed Rank Test. This non-parametric test is suitable for comparing two related samples, matched samples, or repeated measurements on a single sample to assess whether their population mean ranks differ. The Wilcoxon Signed Rank Test does not assume normality, making it a robust choice for our analysis [55].

This test works by ranking the absolute differences between paired observations, assigning ranks, and then computing the sum of ranks for the positive and negative differences. The test statistic $W$ is the smaller of these sums. The p-value is derived from the distribution of $W$ under the null hypothesis that there is no difference between the paired samples.

The Wilcoxon Signed Rank Test is calculated as follows:

- Calculate the differences between paired observations.

- Rank the absolute differences.

- Assign ranks to the differences.

- Compute the sum of ranks for positive and negative differences.

- The test statistic $W$ is the smaller sum of ranks.

- The p-value is obtained from the distribution of $W$.

# 4. Experimental setup

## 4.1 Data preprocessing

We collected the dataset containing information on net demand and consumption in the UK, covering the period from 2020 to 2023. While preparing the data, we apply techniques to handle missing data points, which are represented as NaN or null values within our dataset. To impute the missing values, we calculate the mean $\mu$ and median $\sigma$, that can effectively transform a feature set X from $X = \{x_1, x_2, \ldots, x_n\}$ to $X' = \{x'_1, x'_2, \ldots, x'_n\}$, where $x'_i = x_i$ if $x_i$ is not missing, and $x'_i = \mu$ or $\sigma$ otherwise.

There are three categories of data transformation that are typically utilized in the neural network. In this model, we used linear transformation Eq (16), which scales the data either into (0 to 1) or (-1 to +1) range.

$$\text{MinMaxScaler } X' = \frac{x - \min(x)}{\max(x) - \min(x)} \qquad (16)$$

## 4.2 Data split

In order to account for any differences in data behavior, the dataset is analyzed by classifying it according to temporal patterns, specifically by differentiating between weekdays and weekends. The entire dataset is divided into two parts. 80% of the data was chosen for training, while the remaining 20% was used for validation.

Validation data is further refined by dividing data into four distinct samples (S1, S2, S3, S4), enabling a comprehensive evaluation that enhances the robustness and generalizability of findings.

## 4.3 Preliminary data analysis

The dataset contains a total of 43 features, out of which, after applying GA, we got the four highly correlated features with the target feature for the training for our model. We have done exploratory analysis on these features to get better insights about the data. We separate a single feature on which we wish to make the prediction. In our case, it is Net Demand. We need to scale this feature, so we used the MinMaxScaler library of Python. The dataset did not contain any missing values.

The two-month training graph shows a strong daily cyclical pattern with daytime peaks and nighttime drops, as shown in Fig 5. Energy demand ranges between 17,500 and 35,000 daily, showing no long-term trend. The graph's density shows high-frequency data collection, providing an in-depth view of energy consumption trends with potential outliers from demand drops.

We further illustrated the variations in net energy demand over weekdays and weekends in April and May 2023. It is possible to see a pattern of highs and lows that repeats itself every day, with times when demand rises and times when it falls. Demand changes a lot during the week, with higher levels of consumption during the day and lower levels of demand at night. This repetitive behavior can also be seen on the weekends, but the volume isn't as strong. This could be because people use the system differently when they don't have to work during the week Fig 6.

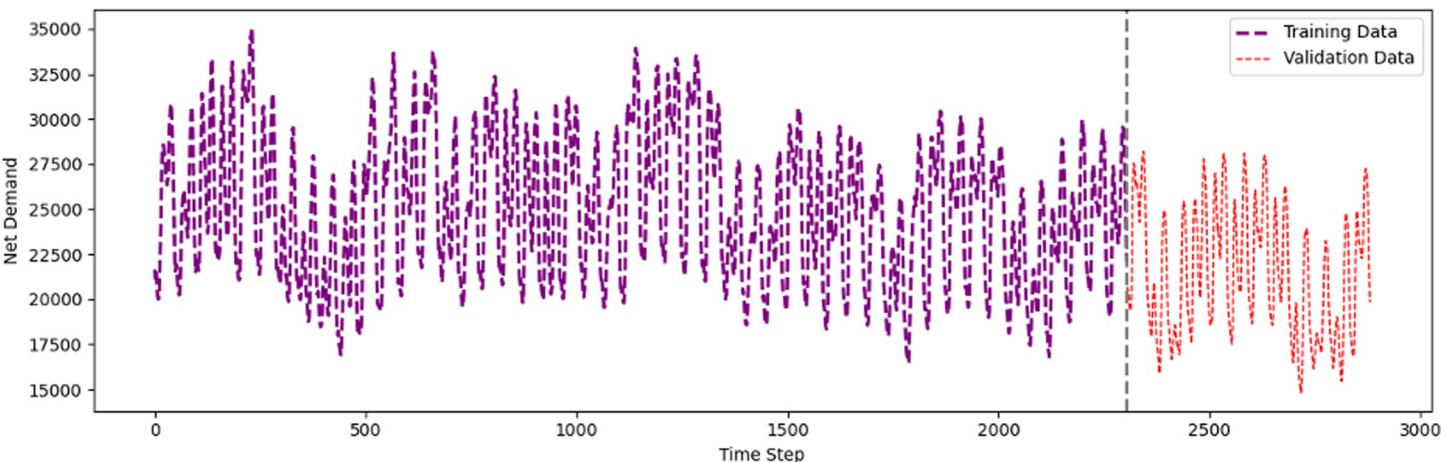

**Fig 5. Pattern of the training and validating data for the last two months.**

Additionally, we used box plots to examine the distribution of energy generation features between weekdays and weekends. The visual representations shown in Fig 7 clearly illustrate a noticeable difference in the core tendencies and variabilities of the two temporal segments. The interquartile ranges shown indicate a greater level of uniformity in energy production patterns on weekdays compared to weekends, especially in the 'GENERATION' and 'FOSSIL' categories. Outliers, identified by the data points located outside the whiskers of the box plots, were present in both categories, indicating occasional deviations from the usual energy production levels.

## 4.4 Stacking ensemble

In this study, we have adopted a stacking approach for the regression problem, considering it the most appropriate, as discussed in [56]. The general structure of this method is illustrated in Fig 8. To define the stacking ensemble scheme more precisely, given P distinct learning algorithms $M_h$, for k = 1 to P, and the data pair $\langle a, b \rangle$ {a = ($a_1$, …., $a_z$) representing the recorded z

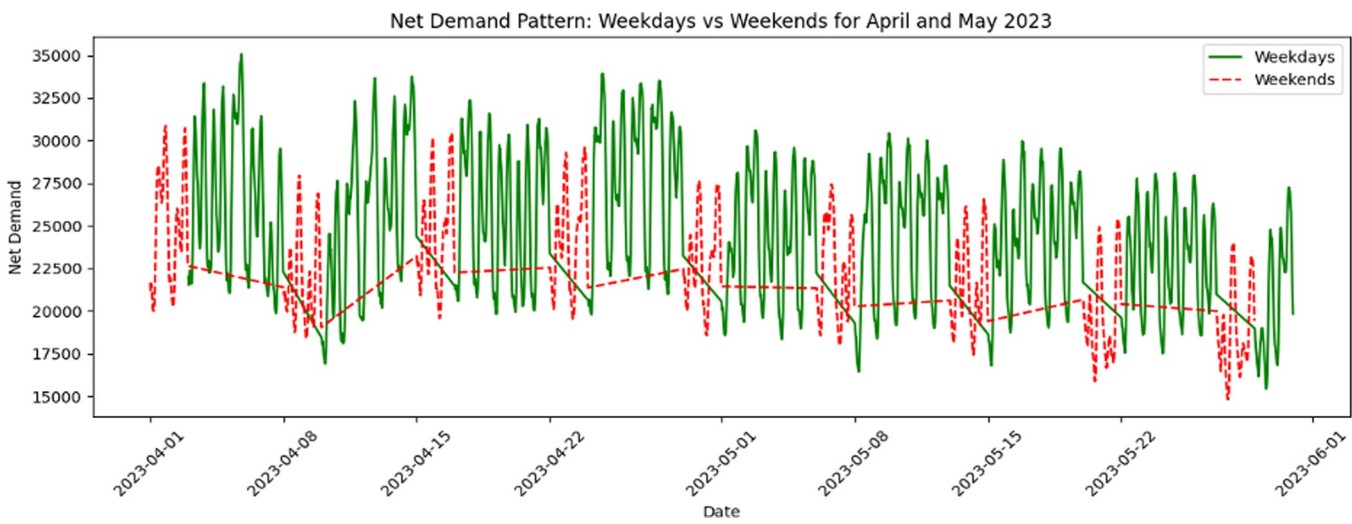

**Fig 6. Weekdays and weekend patterns for net demand.**

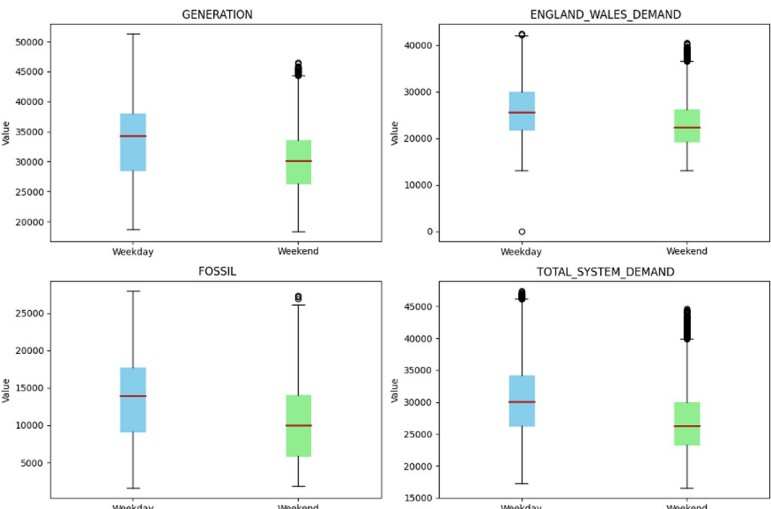

**Fig 7. Box plot the selected feature for weekends and weekdays in the dataset.**

values and b = $(a_{z+1}, …., a_{z+k})$ k values to be predicted}, let $c_{hj}$, (for h = 1 to P, j = 1 to k) be the model generated by $M_h$ to forecast $a_{z+j}$. The function $g_j$, which combines these models for prediction, can be a standard function developed through another algorithm utilizing a machine learning system. The predicted value $\hat{a}_{z+j}$ is then calculated using this method, Eq (17).

$$\hat{a}_{z+j} = g_j\left(d_{1j}, …, d_{Pj}\right) \tag{17}$$

## 4.5 Hyperparameter tuning

In this work, we have employed and optimized various hyperparameters, such as Epoch, which represents a complete iteration of training data. Dropout mitigated the issue of

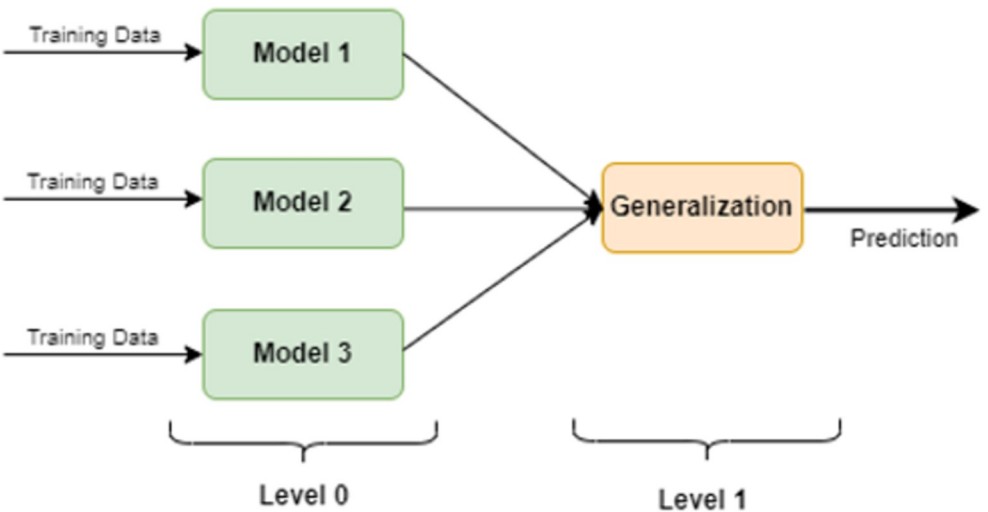

**Fig 8. A basic demonstration of the stacking ensemble technique.**

overfitting. The learning rate refers to the pace at which the model updates its parameters during training. Batch size, on the other hand, represents the number of samples processed by the network at each iteration. The hidden layer, positioned between the input and output layers, determines the number of nodes it should contain. In order to optimize GPU utilization, we have implemented a batch size of 12. Additionally, we have chosen a dropout rate of 0.2 to mitigate overfitting. The dropout rate is 0.2. By removing 20% of the nodes in the hidden layer, we effectively minimize the occurrence of overfitting. The learning rate should not be very low; hence, the learning rate is adjusted to 0.1. After all these adjustments, we achieved a reduction in the mean squared error (MSE) to a level below 0.01, as shown in Fig 9.

## 4.6 Evaluation criteria

The efficacy of our study was evaluated using several variables [57]. The following procedures were used for each, and we have shown in these equations in Eqs (18), (19), (20), and (21)

- *Mean Square Error (MSE)*

$$\text{MSE} = \frac{1}{n} \sum_{i=1}^{n} \left( Y_{\text{Predicted}} - Y_{\text{Actual}} \right)^2 \tag{18}$$

- *Root Mean Square Error (RMSE)*

$$\text{RMSE} = \sqrt{\frac{1}{n} \sum_{i=1}^{n} \left( Y_{\text{Predicted}} - Y_{\text{Actual}} \right)^2} \tag{19}$$

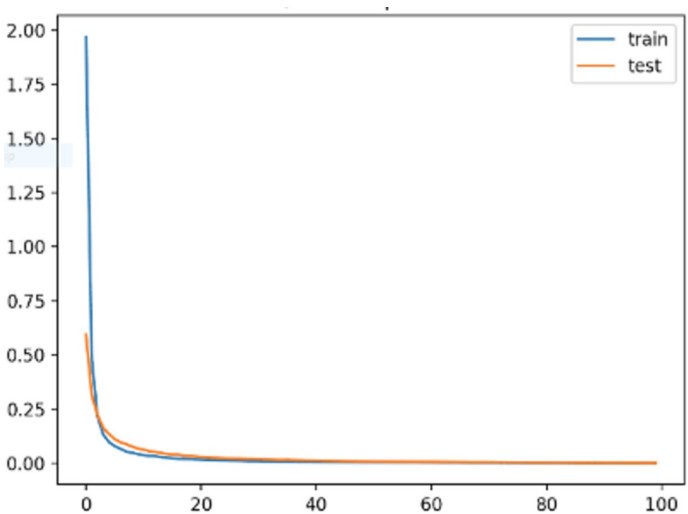

**Fig 9. MSE score plot with epoch.**

- *Mean Absolute Percentage Error (MAPE)*

$$\text{MAPE} = \frac{1}{n} \sum_{1}^{n} \left| \frac{Y_{\text{actual}} - Y_{\text{predicted}}}{Y_{\text{Actual}}} \right| \tag{20}$$

- *Accuracy*

The accuracy of a model is calculated from MAPE.

$$\text{Accuracy} = \left( \frac{1}{n} \sum_{1}^{n} \left| \frac{Y_{\text{actual}} - Y_{\text{predicted}}}{Y_{\text{Actual}}} \right| \right) \tag{21}$$

## 5. Proposed framework

Our proposed method includes a multi-step process that starts with data preparation and culminates with evaluating the predictions. Technically, a common preprocessing formula for data preprocessing and feature selections is presented in Eqs (22) and (23),

- Preprocessing:

$$D_p = \emptyset \left( D \right) \tag{22}$$

Here, D is the original dataset, $D_p$ is the preprocessed dataset, and $\emptyset$ is the preprocessing function that includes data cleaning, transformation, normalization, and temporal feature extraction.

- Feature Selection

$$F_s = \text{GA} \left( D_p \right) \tag{23}$$

Here $F_s$ is the set of selected features, and GA is the Genetic Algorithm applied to $D_p$ for feature selection.

- Base Learners

We deploy three base deep learning models, LSTM, BiLSTM, and GRU, which we have already discussed in section 3.3. The LSTM model is adept at capturing long-term dependencies, the BiLSTM model leverages information from both past and future time points, and the GRU model efficiently models short-term dependencies shown in Eqs (24) – (26).

$$\text{LSTM}: \quad \text{LSTM} \left( F_s \right) = h_t^{LSTM} \tag{24}$$

$$\text{GRU}: \quad \text{GRU} \left( F_s \right) = h_t^{GRU} \tag{25}$$

$$\text{BiLSTM}: \quad \text{BiLSTM} \left( F_s \right) = h_t^{BiLSTM} \tag{26}$$

Here, $h_t$ represents the hidden state at time t.

- Stacking Ensemble

The hidden states of each base learner are combined using a stacking ensemble approach with a linear regression meta-learner presented in Eqs (27) and (28), respectively.

$$\mathrm{H} = \left[ h_t^{LSTM}, h_t^{GRU}, h_t^{BiLSTM} \right] \tag{27}$$

$$\hat{y} = \beta_0 + \beta_1 h_t^{LSTM} + \beta_2 h_t^{GRU} + \beta_3 h_t^{BiLSTM} \tag{28}$$

Here, H is the concatenated hidden states from each base learner, $\hat{y}$ is the predicted energy demand, and $\beta_0, \beta_1, \beta_2, \beta_3$ are the coefficients learned by the meta-learner. When the whole procedure is expressed as an equation, it can be written as in Eq (29)

$$\hat{y} = \beta_0 + \sum_{k=1}^{K} \beta_k \cdot \mathcal{M}_k \left( \mathrm{GA}(\emptyset(D)) \right) \tag{29}$$

Fig 10 demonstrates the entire process of the proposed framework. The model comprises several stages. A thorough preprocessing procedure is applied to the dataset, which includes data cleaning to fix errors, data transformation to organize the information for analysis, variable standardization to make them more comparable, and temporal feature extraction to take

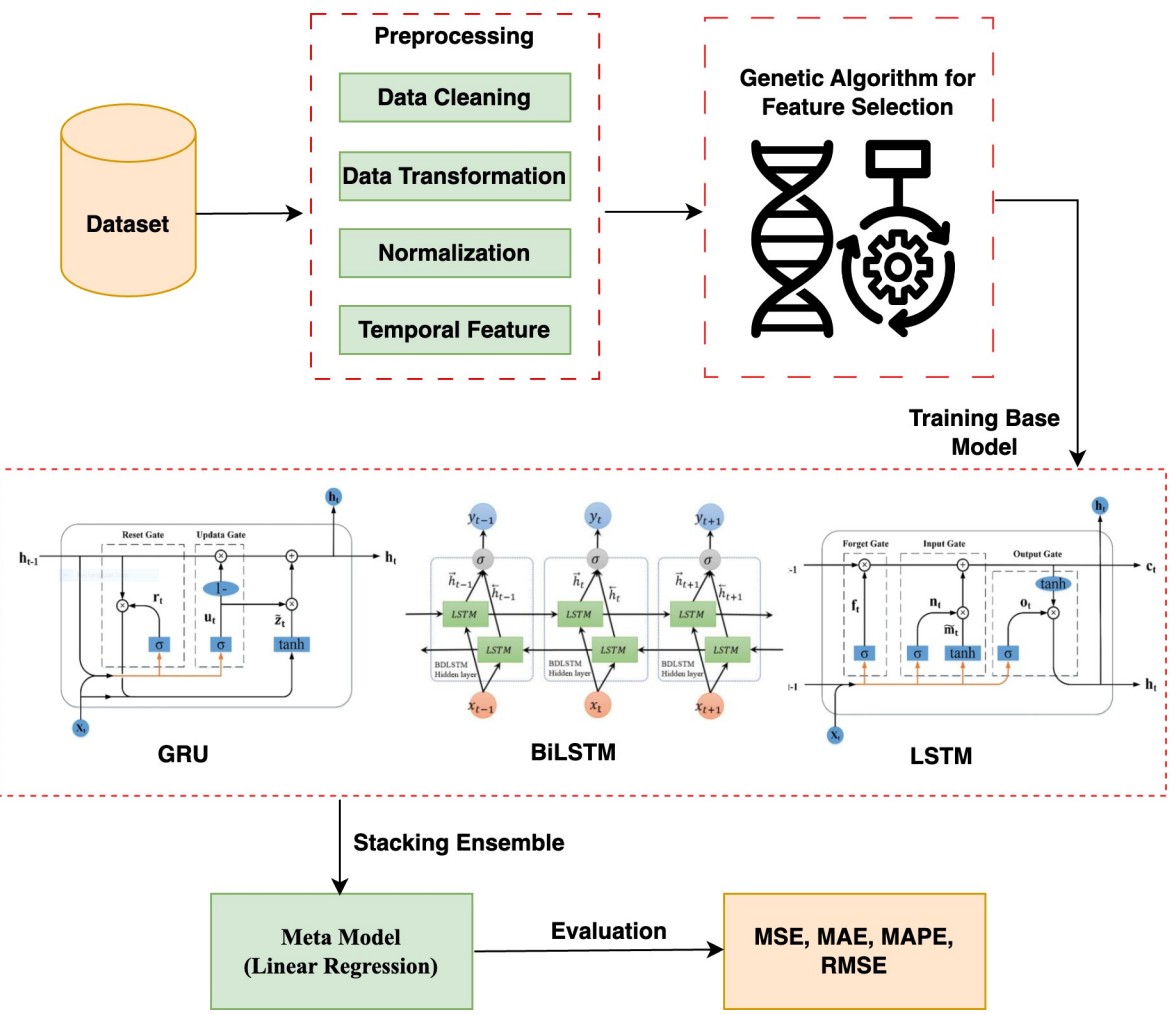

**Fig 10. Overall workflow diagram of the proposed model.**

advantage of trends that change over time. After the preprocessing is completed, a genetic algorithm is then used for feature selection. This computational evolution approach uses mutation and crossover to enhance the prediction skills of the feature set population, mimicking natural selection. The whole GA process is shown in Fig 1. We evaluate the fitness of each feature set based on how well it predicts energy use.

In the model implementation phase, three separate recurrent neural networks—GRU, BiLSTM, and LSTM—are trained as the base models using the features that GA selected. We used a stacking ensemble approach to best use each neural network's capabilities. A meta-model is trained in this ensemble to merge the forecasts of the GRU, BiLSTM, and LSTM basis models into a single, more precise prediction. In the last stage, statistical matrices such as MSE, MAE, MAPE, and RMSE are used to measure the accuracy of forecasts and conduct a thorough evaluation of this composite model's performance.

## 6. Results and discussion

### 6.1 Result

The performance of the proposed model was evaluated through multiple metrics, namely RMSE, MAE, and MAPE. Detailed mathematical formulations for these evaluation metrics are provided in Section 4.6.

In the dataset analyzed for this study, two distinct energy demand patterns were identified, as shown in Fig 6. The first pattern is associated with weekends, where a notable decrease in energy demand is observed. In contrast, the second pattern corresponds to weekdays, characterized by a consistent increase in energy demand. The data was systematically categorized according to these patterns, and the model was then trained to handle them effectively. The results of this comparative analysis are displayed in Fig 11 for the weekday pattern and in Fig 12 for the weekend pattern.

After training the architecture presented in this study using the data patterns illustrated in Fig 6, we first tested the model on the training dataset. The outcomes of this test are detailed in Table 2. To further our objective, we then evaluated the model's performance using a hidden validation dataset from the energy demand data. The results of this evaluation are shown in Table 3.

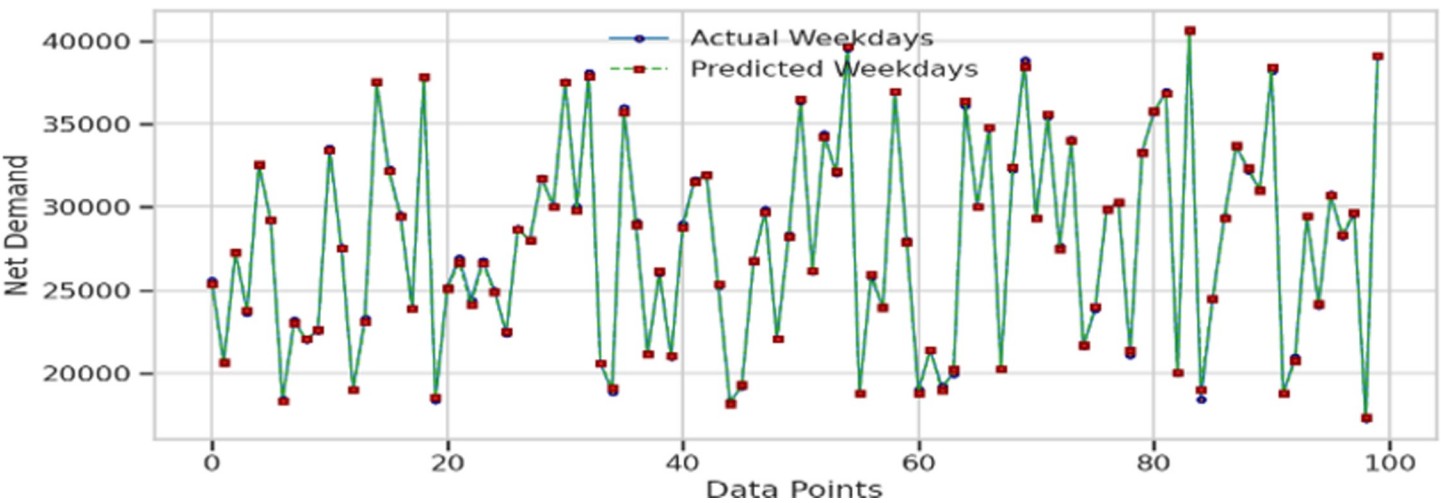

**Fig 11. Plot for whole validation data for weekdays training.**

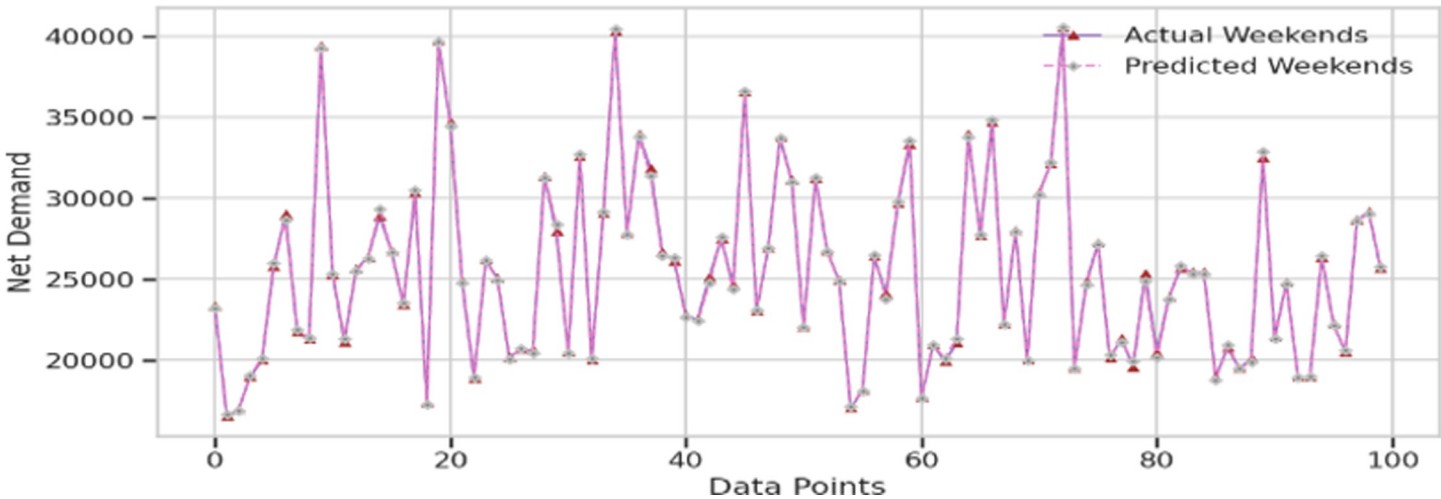

**Fig 12. Plot for whole validation data for weekend training.**

An additional performance assessment was also conducted by dividing the entire validation dataset into four distinct samples. The model's performance was then re-evaluated on each of these samples separately. The outcomes of these evaluations are documented in Fig 13 and Table 4.

In the final analysis, individual models using LSTM, GRU, and BiLSTM architectures were developed for the identified demand patterns. Their comparative performance is illustrated in Fig 14.

In Table 4, we provide a comprehensive evaluation, presenting the overall performance scores of all models across the four validation samples.

Deep learning models are inherently stochastic, which means that they can produce different results on different executions. To demonstrate the reliability of our proposed model, we executed the entire program ten times. For each run, we recorded the error metrics, as shown in Tables 5 and 6.

## 6.2 Wilcoxon signed-rank test

To determine if the differences in performance metrics across different runs were statistically significant, we applied the Wilcoxon Signed Rank Test. The results are summarized in Table 7.

The p-values indicate whether there is a significant difference between the models' performances over multiple runs. A p-value less than 0.05 typically indicates a significant difference. In this case, the p-values for all comparisons (Ensemble vs. LSTM, Ensemble vs. BiLSTM, and Ensemble vs. GRU) across all metrics (MSE, MAE, RMSE, MAPE) are 0.001953125, which is significantly less than the 0.05 threshold. This demonstrates that the ensemble model

**Table 2. Overall performance evaluation of the training data.**

| Model | Weekdays | | | Weekends | | |
|---|---|---|---|---|---|---|
| | RMSE | MAPE (%) | MAE | RMSE | MAPE (%) | MAE |
| LSTM | 137.24 | 0.40 | 108.36 | 172.36 | 0.54 | 134.28 |
| BiLSTM | 130.26 | 0.37 | 101.24 | 139.24 | 0.43 | 102.69 |
| GRU | 132.65 | 0.38 | 105.46 | 165.24 | 0.56 | 125.98 |
| **Stacking-GA** | **121.24** | **0.35** | **96.34** | **1.32.64** | **0.37** | **102.61** |

**Table 3. Overall performance evaluation on the validation data.**

|  | Weekdays | | | Weekends | | |
|---|---|---|---|---|---|---|
| **Model** | **RMSE** | **MAPE (%)** | **MAE** | **RMSE** | **MAPE (%)** | **MAE** |
| LSTM | 146.18 | 0.42 | 114.36 | 180.34 | 0.59 | 142.13 |
| BiLSTM | 135.53 | 0.38 | 105.67 | 142.18 | 0.45 | 109.56 |
| GRU | 140.26 | 0.39 | 107.45 | 178.99 | 0.59 | 140.73 |
| **Stacking-GA** | **130.61** | **0.38** | **99.41** | **137.41** | **0.42** | **105.67** |

consistently shows a statistically significant improvement in performance over the individual models.

The prediction accuracy criteria for assessment are schematically shown in Fig 15. Most of the error lies around zero. Therefore, mathematical findings confirm the predicted accuracy and effectiveness of the proposed approach.

## 6.3 Discussion

The findings indicate that the Stacking-GA model consistently achieves better results compared to the separate deep learning models in all criteria during both weekdays and weekends. In Sample S1 on weekdays, the Stacking-GA model had the lowest MAE (494.62) for weekdays Fig 16(A), and MAE (534.25) for weekends Fig 16(B), which indicates its improved predictive accuracy compared to the solo models. Similarly, the observed trend is stable in all samples, suggesting that the ensemble technique successfully captures the fundamental patterns in energy consumption.

The RMSE values, which indicate the level of volatility in the predictions, drop in the following order: LSTM, GRU, BiLSTM, and Stacking-GA. This further highlights the resilience of the ensemble approach. Furthermore, the MAPE shown in Fig 17(A) and 17(B), and RMSE shown in Fig 18(A) and 18(B) provide additional evidence supporting the enhanced precision and dependability of the Stacking-GA model. The performance difference amongst the models is particularly evident on weekends, possibly indicating the enhanced challenge of predicting energy demand during these periods due to less predictable consumption patterns.

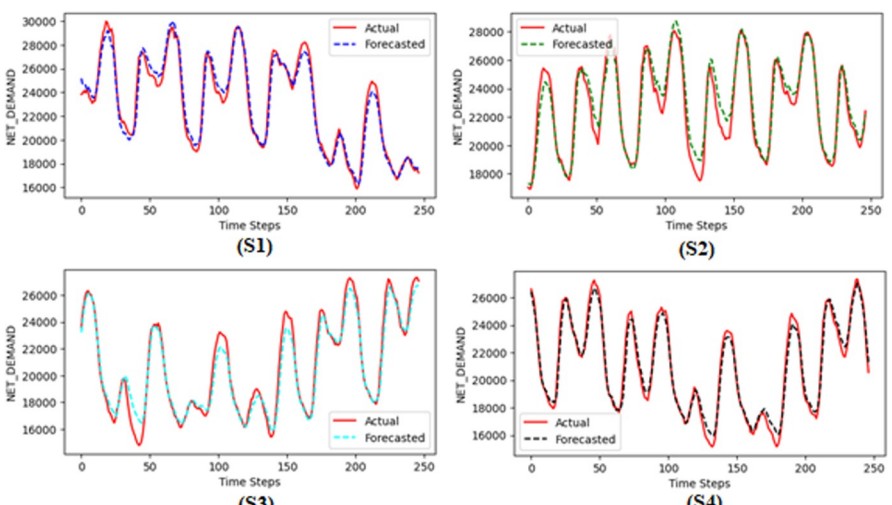

**Fig 13.** Actual vs predicted graph for all four samples (S1, S2, S3, S4).

**Table 4. Performance evaluation of four samples on the validation data.**

| Validation samples | | Weekdays | | | | Weekends | | | |
|---|---|---|---|---|---|---|---|---|---|
| | | LSTM | GRU | BiLSTM | Stacking-GA | LSTM | GRU | BiLSTM | Stacking - GA |
| S1 | RMSE | 758.36 | 685.49 | 624.25 | **599.49** | 705.36 | 792.34 | 786.32 | **742.56** |
| | MAPE(%) | 2.68 | 2.48 | 2.36 | **2.11** | 2.56 | 2.49 | 2.42 | **2.32** |
| | MAE | 596.35 | 558.36 | 515.69 | **494.62** | 674.21 | 624.35 | 598.47 | **534.25** |
| S2 | RMSE | 869.35 | 832.14 | 795.65 | **761.34** | 996.05 | 916.74 | 887.14 | **846.32** |
| | MAPE(%) | 2.98 | 2.91 | 2.85 | **2.67** | 3.74 | 3.54 | 3.10 | **2.89** |
| | MAE | 745.71 | 634.85 | 614.96 | **592.89** | 704.52 | 690.36 | 671.52 | **614.63** |
| S3 | RMSE | 799.54 | 782.65 | 715.54 | **684.35** | 893.21 | 785.34 | 740.96 | **732.26** |
| | MAPE(%) | 3.45 | 2.96 | 2.54 | **2.43** | 2.79 | 2.74 | 2.63 | **2.54** |
| | MAE | 745.36 | 648.25 | 540.12 | **493.23** | 741.36 | 647.98 | 569.38 | **536.45** |
| S4 | RMSE | 596.34 | 569.47 | 554.25 | **514.49** | 796.54 | 690.68 | 649.25 | **589.62** |
| | MAPE(%) | 2.84 | 2.65 | 2.12 | **2.02** | 2.45 | 2.39 | 2.36 | **2.13** |
| | MAE | 541.36 | 470.85 | 430.24 | **415.50** | 659.25 | 554.14 | 490.65 | **460.65** |

In summary, the Stacking-GA model demonstrates a significant improvement in prediction accuracy compared to individual deep learning models such as LSTM, BiLSTM, and GRU. Due to the stochastic nature of these models, we conducted repeated simulations, running each model ten times with different random seeds and recording the performance metrics (MSE, MAE, RMSE, MAPE) for each run. To ensure the reliability of our results, we used the Wilcoxon Signed Rank Test to compare the performance metrics across the different runs.

The Wilcoxon Signed Rank Test results, with p-values of 0.001953125 for all comparisons, indicate a statistically significant improvement in the performance of the ensemble model over the individual models across all metrics. The enhanced performance in energy demand forecasting is likely attributed to the combination of feature selection, ensemble learning, and the incorporation of a Genetic Algorithm.

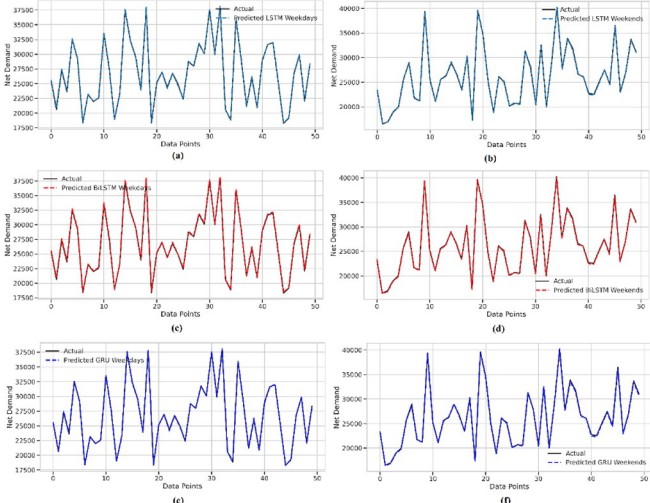

**Fig 14. Actual vs predicted net demand for all three models.** LSTM - (a) for weekend. (a') for weekdays. BiLSTM - (b) for weekdays, (b') for weekend. GRU - (c) for weekdays, (c') for weekend.

**Table 5. Overall performance metrics across ten runs on weekdays dataset.**

| Model Error | No of Trails | 1st | 2nd | 3rd | 4th | 5th | 6th | 7th | 8th | 9th | 10th |
|---|---|---|---|---|---|---|---|---|---|---|---|
| RMSE | LSTM | 148.3 | 145.2 | 147.9 | 151.8 | 146.8 | 144.98 | 141.8 | 145.3 | 143.7 | 150.6 |
| | BiLSTM | 138.5 | 141.2 | 139.3 | 137.5 | 134.9 | 133.9 | 139.4 | 132.5 | 137.9 | 136.4 |
| | GRU | 145.2 | 147.2 | 138.9 | 145.7 | 143.8 | 147.9 | 141.5 | 149.6 | 146.8 | 157.6 |
| | **Stacking-GA** | 130.8 | 131.5 | 136.4 | 140.8 | 136.9 | 145.2 | 135.6 | 147.6 | 142.8 | 150.6 |
| MAPE % | LSTM | 0.42 | 0.42 | 0.41 | 0.43 | 0.41 | 0.44 | 0.43 | 0.45 | 0.42 | 0.41 |
| | BiLSTM | 0.38 | 0.34 | 0.37 | 0.36 | 0.39 | 0.38 | 0.40 | 0.39 | 0.37 | 0.35 |
| | GRU | 0.40 | 0.41 | 0.43 | 0.37 | 0.36 | 0.39 | 0.38 | 0.41 | 0.42 | 0.34 |
| | **Stacking-GA** | 0.35 | 0.36 | 0.39 | 0.34 | 0.33 | 0.35 | 0.37 | 0.39 | 0.41 | 0.32 |
| MAE | LSTM | 115.3 | 118.6 | 115.2 | 114.5 | 116.9 | 112.3 | 119.6 | 121.6 | 117.5 | 116.9 |
| | BiLSTM | 106.8 | 109.9 | 112.5 | 108.6 | 110.5 | 107.6 | 115.2 | 118.8 | 116.3 | 113.2 |
| | GRU | 107.6 | 108.3 | 111.6 | 105.7 | 109.4 | 105.3 | 112.1 | 113.9 | 115.7 | 110.0 |
| | **Stacking-GA** | 99.6 | 101.2 | 98.6 | 97.3 | 101.4 | 102.9 | 99.3 | 108.2 | 104.5 | 98.7 |

## 6.4 Implications and potential application

The ensemble deep learning framework proposed in this study has significant implications and potential applications in real-world energy demand forecasting scenarios. It provides accurate and robust forecasts, which are crucial for various stakeholders in the energy section.

**6.4.1 Enhanced forecasting accuracy.** The ensemble framework, which combines the LSTM, BiLST, and GRU models, improves the accuracy of energy demand predictions. Better accuracy is crucial for utility firms, grid operators, and policymakers to make well-informed choices about energy generation, distribution, and management.

**6.4.2 Adaptability to different time horizons.** Our method is adaptable to various forecasting horizons, whether short-term, medium-term, or long-term. This adaptability makes it suitable for different applications, such as day-ahead forecasting for grid operation, week-ahead forecasting for maintenance scheduling, and long-term forecasting for infrastructure planning.

**6.4.3 Handling complex and high-dimensional data.** The use of a GA for feature selection allows the framework to handle complex and high-dimensional data efficiently. This

**Table 6. Overall performance metrics across ten runs on the weekend dataset.**

| Model Error | No of Trails | 1st | 2nd | 3rd | 4th | 5th | 6th | 7th | 8th | 9th | 10th |
|---|---|---|---|---|---|---|---|---|---|---|---|
| RMSE | LSTM | 180.6 | 182.6 | 185.2 | 184.6 | 181.4 | 196.3 | 189.7 | 184.5 | 180.2 | 186.1 |
| | BiLSTM | 142.5 | 145.6 | 148.7 | 151.4 | 150.3 | 149.6 | 148.5 | 151.9 | 150.4 | 152.9 |
| | GRU | 175.2 | 179.3 | 174.5 | 178.9 | 180.6 | 181.6 | 170.5 | 171.6 | 177.2 | 178.6 |
| | **Stacking-GA** | 135.2 | 136.5 | 134.2 | 140.6 | 142.7 | 143.9 | 138.7 | 139.6 | 145.7 | 149.3 |
| MAPE % | LSTM | 0.59 | 0.58 | 0.60 | 0.59 | 0.61 | 0.62 | 0.59 | 0.60 | 0.58 | 0.62 |
| | BiLSTM | 0.45 | 0.47 | 0.46 | 0.45 | 0.44 | 0.53 | 0.49 | 0.46 | 0.47 | 0.48 |
| | GRU | 0.59 | 0.57 | 0.59 | 0.58 | 0.60 | 0.57 | 0.60 | 0.61 | 0.59 | 0.58 |
| | **Stacking-GA** | 0.42 | 0.41 | 0.39 | 0.42 | 0.40 | 0.41 | 0.45 | 0.43 | 0.39 | 0.44 |
| MAE | LSTM | 142.3 | 144.5 | 143.1 | 145.6 | 148.2 | 140.7 | 143.9 | 140.9 | 144.5 | 147.6 |
| | BiLSTM | 109.6 | 109.8 | 110.2 | 112.4 | 108.6 | 111.3 | 115.2 | 112.5 | 116.3 | 113.2 |
| | GRU | 140.3 | 138.9 | 142.7 | 139.6 | 141.5 | 142.7 | 144.5 | 143.2 | 140.8 | 141.3 |
| | **Stacking-GA** | 105.9 | 102.3 | 106.4 | 104.7 | 103.2 | 101.9 | 108.9 | 104.2 | 107.1 | 106.9 |

**Table 7. Wilcoxon signed-rank test results.**

| Comparison | Statistic | p-value |
|---|---|---|
| Stacking-GA vs LSTM | 0.0 | 0.001953125 |
| Stacking-GA vs BiLSTM | 0.0 | 0.001953125 |
| Stacking-GA vs GRU | 0.0 | 0.001953125 |

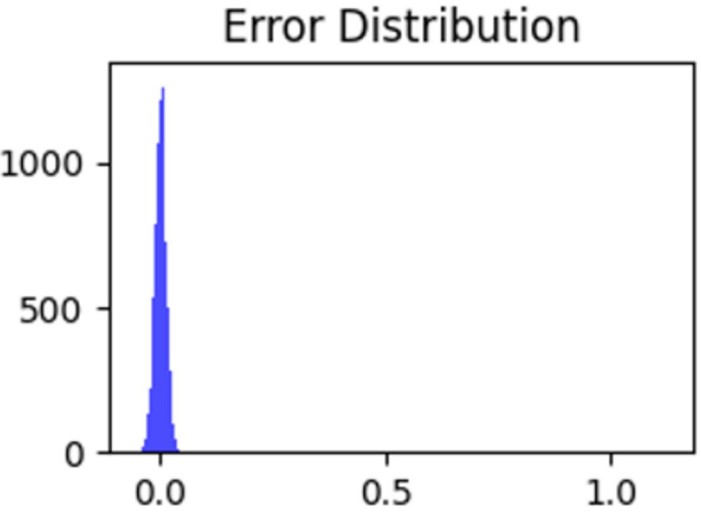

**Fig 15. Error distribution.**

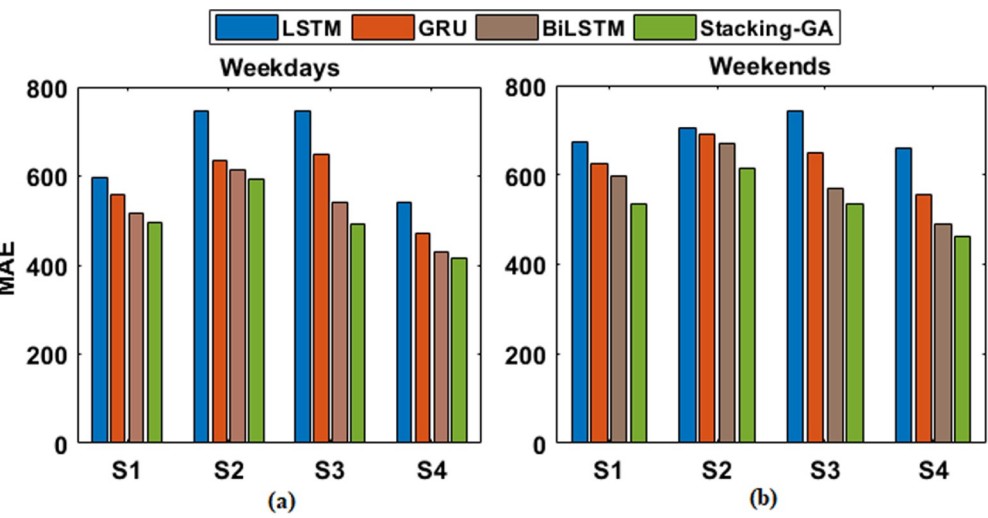

**Fig 16.** MAE score of LSTM, GRU, BiLSM, and Stacking-GA based model for both (a) weekdays and (b) weekends.

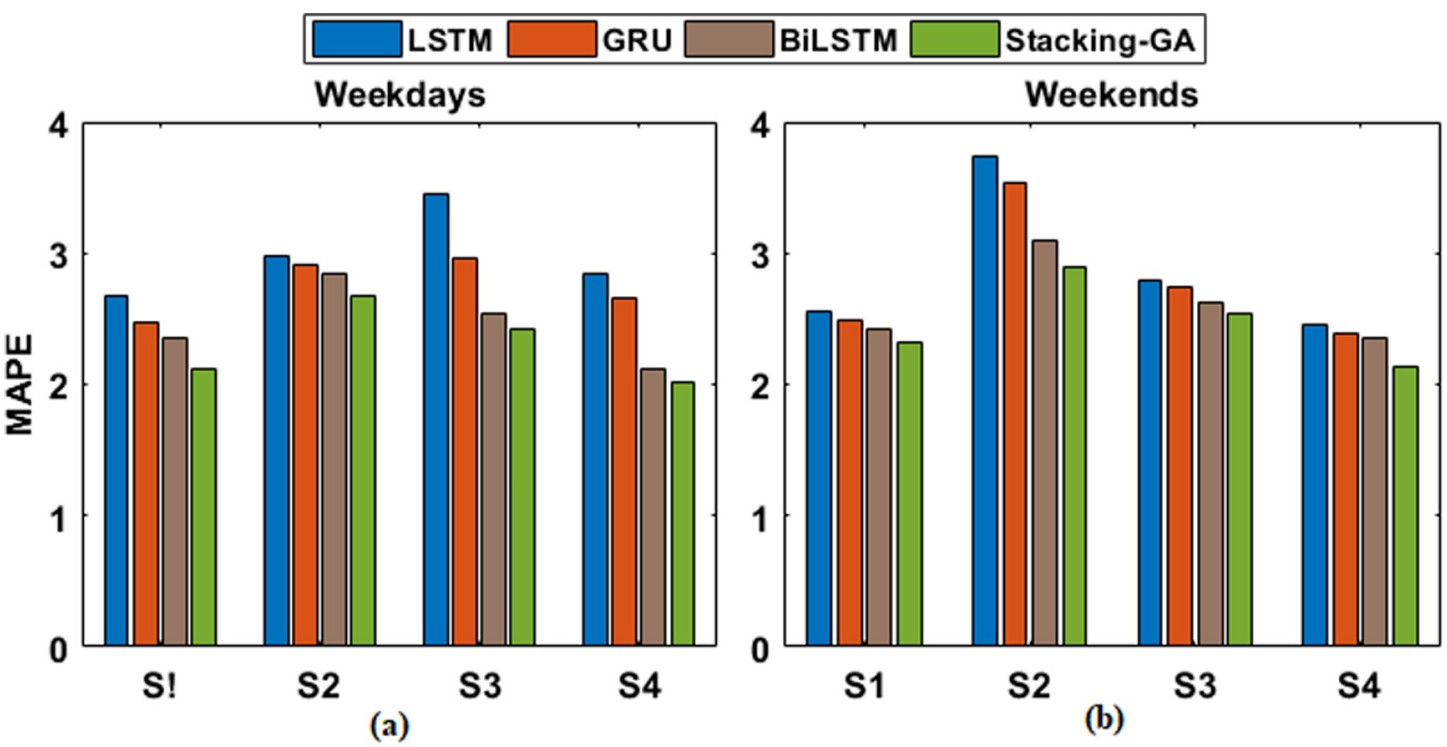

**Fig 17.** MAPE score of LSTM, GRU, BiLSM, and Stacking-GA based model for both (a) weekdays and (b) weekends.

capability is particularly useful in scenarios where large amounts of data from smart meters, weather stations, and other sources need to be processed to generate accurate forecasts.

**6.4.4 Real-time forecasting and smart grids.** With developments in real-time data collecting and processing technology, our ensemble framework may be integrated into smart grid systems to give real-time energy demand forecasts. This has the potential to increase energy

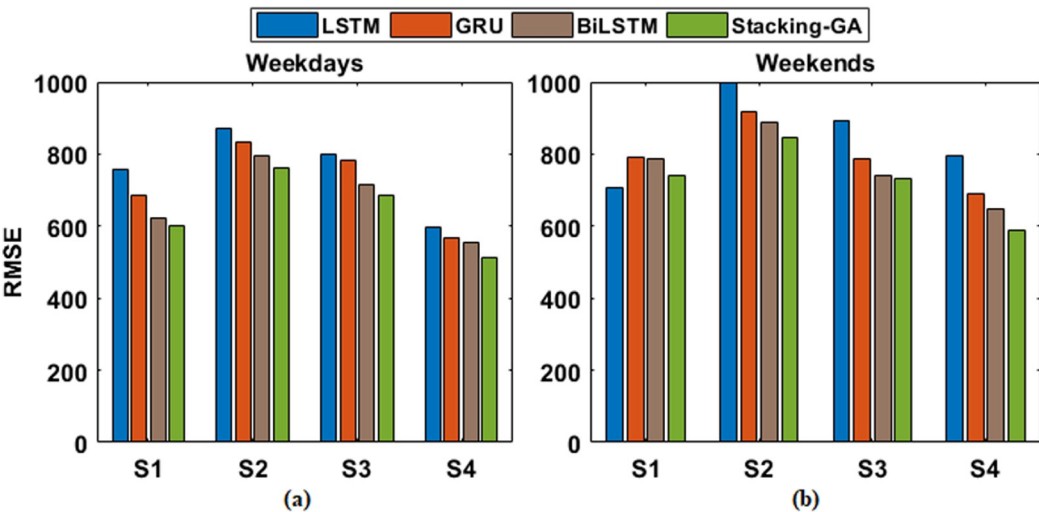

**Fig 18.** RMSE score of LSTM, GRU, BiLSM, and Stacking-GA based model for both (a) weekdays and (b) weekends.

distribution efficiency and reliability while also making it easier to integrate distributed energy resources.

## 7. Conclusions

This research presents a significant advancement in energy demand forecasting by introducing an innovative stacking ensemble approach. The integration of LSTM, GRU, and BiLSTM networks, underpinned by a genetic algorithm for feature selection, marks a novel methodology in addressing the complexities of energy demand prediction. We have also utilized the Wilcoxon Signed Rank test for statistical validation and ran the simulations ten times to ensure our results hold up. The performance evaluation was conducted using three key metrics: RMSE, MAPE, and MAE. To ensure the model's robustness against the variability in energy consumption patterns, the data was segmented into weekday and weekend categories for analysis. The results from the validation data highlight the model's exceptional precision, with a weekday performance marked by an RMSE of 130.6, a MAPE of 0.38%, and an MAE of 99.41. For weekend projections, the model maintained its accuracy, recording an RMSE of 137.41, a MAPE of 0.42%, and an MAE of 105.67. This level of precision indicates the model's effectiveness in capturing the complexities of energy demand patterns. Additionally, The use of a genetic algorithm for feature selection has proven to be a key factor in the model's success. It efficiently identifies the most influential predictors, improving the model's performance. Furthermore, The stacking-based ensemble model, integrating multiple deep learning techniques, showcases a robust framework that outperforms traditional single-model approaches in forecasting accuracy. The study not only contributes to the theoretical understanding of feature selection in machine learning but also offers practical implications for energy analysts and policymakers. Enhancing the accuracy of energy demand forecasts aids in efficient energy management and planning, which is crucial in the context of growing energy needs and sustainability challenges.

## Author Contributions

**Conceptualization:** Mohd Sakib, Tamanna Siddiqui.

**Data curation:** Mohd Sakib.

**Formal analysis:** Mohd Sakib.

**Investigation:** Mohd Sakib.

**Methodology:** Mohd Sakib, Tamanna Siddiqui.

**Resources:** Mohd Sakib, Tamanna Siddiqui, Reemiah Muneer Alotaibi, Nouf Mohammad Alshareef, Mohammad Zunnun Khan.

**Software:** Mohd Sakib.

**Supervision:** Tamanna Siddiqui, Suhel Mustajab.

**Validation:** Mohd Sakib, Suhel Mustajab.

**Visualization:** Mohd Sakib.

**Writing – original draft:** Mohd Sakib.

**Writing – review & editing:** Tamanna Siddiqui, Suhel Mustajab, Reemiah Muneer Alotaibi, Nouf Mohammad Alshareef, Mohammad Zunnun Khan.

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
