## [Decision Letter · Decision Letter 0]

5 Jun 2024

PONE-D-24-12727An Ensemble Deep Learning Framework for Energy Demand Forecasting Using Genetic Algorithm-Based Feature SelectionPLOS ONE

Dear Dr. Siddiqui,

Thank you for submitting your manuscript to PLOS ONE. After careful consideration, we feel that it has merit but does not fully meet PLOS ONE’s publication criteria as it currently stands. Therefore, we invite you to submit a revised version of the manuscript that addresses the points raised during the review process.

The author have employed stochastic DL models (LSTM, BiLSTM, GRU) and a stochastic method GA. Therefore, it is suggested to repeat the simulations at least 10 times and apply non-parametric statistical tests like Wilcoxon Signed Rank Test on the obtained results to draw decisive conclusions. Otherwise the conclusions might not be reliable.

We look forward to receiving your revised manuscript.

Kind regards,

Sibarama Panigrahi, PhD

Academic Editor

PLOS ONE

“No”

4. PLOS requires an ORCID iD for the corresponding author in Editorial Manager on papers submitted after December 6th, 2016. Please ensure that you have an ORCID iD and that it is validated in Editorial Manager. To do this, go to ‘Update my Information’ (in the upper left-hand corner of the main menu), and click on the Fetch/Validate link next to the ORCID field. This will take you to the ORCID site and allow you to create a new iD or authenticate a pre-existing iD in Editorial Manager. Please see the following video for instructions on linking an ORCID iD to your Editorial Manager account: https://www.youtube.com/watch?v=_xcclfuvtxQ.

Additional Editor Comments:

The authors have employed stochastic DL models (LSTM, BiLSTM, GRU) and a stochastic method GA. Therefore, it is suggested to repeat the simulations at least 10 times and apply non-parametric statistical tests like Wilcoxon Signed Rank Test on the obtained results to draw decisive conclusions. Otherwise the conclusions might not be reliable.

Reviewers' comments:

Reviewer's Responses to Questions

**Comments to the Author**

1. Is the manuscript technically sound, and do the data support the conclusions?

Reviewer #1: Yes

Reviewer #2: Yes

2. Has the statistical analysis been performed appropriately and rigorously? 

Reviewer #1: Yes

Reviewer #2: Yes

3. Have the authors made all data underlying the findings in their manuscript fully available?

Reviewer #1: Yes

Reviewer #2: Yes

4. Is the manuscript presented in an intelligible fashion and written in standard English?

Reviewer #1: Yes

Reviewer #2: Yes

5. Review Comments to the Author

Reviewer #1: 1. Check for grammar and puntuation

2. 101 ARIMA is written twice

3. number the equations

4. 181. what does se meams.

5. 221. denine U

6. verify the results in table 2.

7. verify the results in table 4.

Reviewer #2: The following points need to be addressed by the reviewers:

1. Mention the objectives in the abstract.

2. How does GA based feature selection contribute to the performance of the energy demand forecasting model?

3. Elaborate the types of deep learning models that are integrated into the ensemble framework proposed in the paper?

4. All the equations must be numbered and cited in the text.

5. Describe the process of feature selection in details.

6. What are the key challenges addressed by using the proposed method?

7. Discuss the implications and potential applications of the ensemble deep learning framework in real-world energy demand forecasting scenarios.

6. PLOS authors have the option to publish the peer review history of their article (what does this mean?). If published, this will include your full peer review and any attached files.

Reviewer #1: **Yes: **Kishore Kumar Sahu, School of Computer Science, VSSUT, Burla.

Reviewer #2: **Yes: **Dr. Manoj Kumar Kar

---

## [Decision Letter · Decision Letter 1]

23 Aug 2024

PONE-D-24-12727R1An Ensemble Deep Learning Framework for Energy Demand Forecasting Using Genetic Algorithm-Based Feature SelectionPLOS ONE

Dear Dr. Siddiqui,

Thank you for submitting your manuscript to PLOS ONE. After careful consideration, we feel that it has merit but does not fully meet PLOS ONE’s publication criteria as it currently stands. Therefore, we invite you to submit a revised version of the manuscript that addresses the points raised during the review process.

**ACADEMIC EDITOR: **The authors are suggested to carefully go through each sentence of the manuscript and improve the linguistic quality of the manuscript.==============================

We look forward to receiving your revised manuscript.

Kind regards,

Sibarama Panigrahi, PhD

Academic Editor

PLOS ONE

Journal Requirements:

Additional Editor Comments:

The authors are suggested to carefully go through each sentence of the manuscript and improve the linguistic quality of the manuscript.

Reviewers' comments:

Reviewer's Responses to Questions

**Comments to the Author**

1. If the authors have adequately addressed your comments raised in a previous round of review and you feel that this manuscript is now acceptable for publication, you may indicate that here to bypass the “Comments to the Author” section, enter your conflict of interest statement in the “Confidential to Editor” section, and submit your "Accept" recommendation.

Reviewer #1: All comments have been addressed

Reviewer #2: All comments have been addressed

2. Is the manuscript technically sound, and do the data support the conclusions?

Reviewer #1: Yes

Reviewer #2: Yes

3. Has the statistical analysis been performed appropriately and rigorously? 

Reviewer #1: Yes

Reviewer #2: Yes

4. Have the authors made all data underlying the findings in their manuscript fully available?

Reviewer #1: Yes

Reviewer #2: Yes

5. Is the manuscript presented in an intelligible fashion and written in standard English?

Reviewer #1: Yes

Reviewer #2: Yes

6. Review Comments to the Author

Reviewer #1: 1. Check for grammar and punctuation YES

2. 101 ARIMA is written twice YES

3. number the equations YES

4. 181. what does se means. YES

5. 221. define U YES

6. verify the results in table 2. YES

7. verify the results in table 4. YES

All the comments have been resolved

Reviewer #2: The manuscript may be considered in its present form as all the queries are now addressed by the authors.

7. PLOS authors have the option to publish the peer review history of their article (what does this mean?). If published, this will include your full peer review and any attached files.

Reviewer #1: **Yes: **Dr. Kishore Kumar Sahu

Reviewer #2: **Yes: **Manoj Kumar Kar

---

## [Author Response · Author response to Decision Letter 1]

1 Sep 2024

Dear Editors and Reviewers,

We would like to express our sincere gratitude to the editors and reviewers for their thorough examination and insightful feedback on our manuscript. 

In response to the reviews, we have thoroughly revised our manuscript, taking into careful consideration every comment and suggestion provided. Revised portions are highlighted in the manuscript in yellow. The main corrections in the paper and the responses to the reviewer's comments are listed below.

Response to the Editor

Comment: The authors are suggested to carefully go through each sentence of the manuscript and improve the linguistic quality of the manuscript.

Response: In response to the editor's suggestion, we enlisted the help of a native English speaker to thoroughly revise the manuscript. We carefully reviewed each sentence and made several revisions to enhance its linguistic quality. The changes included both minor adjustments and more substantial edits.

 First, we focused on punctuation and formatting, ensuring consistency and clarity throughout the text. This involved adding or removing spaces before and after punctuation marks, as well as making necessary adjustments to commas, periods, and other punctuation for improved readability.

 Second, we refined the wording of numerous sentences and phrases to improve clarity and correct grammatical errors. For example, we revised sentences to ensure they conveyed the intended meaning more effectively and corrected any awkward phrasing or linguistic inaccuracies.

 Additionally, we made structural edits to some parts of the manuscript. This involved reorganizing certain sentences or sections to improve the overall flow and coherence of the text. Furthermore, we removed redundant lines and unnecessary words to streamline the content and eliminate any repetition. This helped to sharpen the focus of the manuscript and enhance its overall readability.

Journal’s Requirements

In compliance with the journal's guidelines, we have thoroughly reviewed the reference list to ensure its completeness and accuracy. As part of this process, we carefully checked for any retracted papers and removed them where necessary. Additionally, we verified the DOI information for all references and have added the DOI to any entries where it was previously missing. All necessary corrections and updates to the references have been made to ensure they meet the required standards and align with the journal's requirements. The references that were corrected are highlighted in yellow and are presented below

[4] G. Franco and A. H. Sanstad, “Climate change and electricity demand in California,” Clim Change, vol. 87, no. 1, pp. 139–151, 2008, doi: 10.1007/s10584-007-9364-y.

[5] S. Noureen, S. Atique, V. Roy, and S. Bayne, “Analysis and application of seasonal ARIMA model in Energy Demand Forecasting: A case study of small scale agricultural load,” Midwest Symposium on Circuits and Systems, vol. 2019-Augus, pp. 521–524, 2019, doi: 10.1109/MWSCAS.2019.8885349.

[11] M. Q. Raza and A. Khosravi, “A review on artificial intelligence based load demand forecasting techniques for smart grid and buildings,” Renewable and Sustainable Energy Reviews, vol. 50, pp. 1352–1372, 2015, doi: 10.1016/j.rser.2015.04.065.

[22] Z. Ullah, F. Al-Turjman, L. Mostarda, and R. Gagliardi, “Applications of Artificial Intelligence and Machine learning in smart cities,” Comput Commun, vol. 154, pp. 313–323, 2020, doi: 10.1016/j.comcom.2020.02.069.

[23] W. Xu, H. Peng, X. Zeng, F. Zhou, X. Tian, and X. Peng, “A hybrid modelling method for time series forecasting based on a linear regression model and deep learning,” Applied Intelligence, vol. 49, no. 8, pp. 3002–3015, 2019, doi: 10.1007/s10489-019-01426-3.

[24] F. Mohammad and Y. C. Kim, “Energy load forecasting model based on deep neural networks for smart grids,” International Journal of System Assurance Engineering and Management, vol. 11, no. 4, pp. 824–834, 2020, doi: 10.1007/s13198-019-00884-9.

[25] T. Ahmad, H. Zhang, and B. Yan, “A review on renewable energy and electricity requirement forecasting models for smart grid and buildings,” Sustain Cities Soc, vol. 55, p. 102052, 2020. doi: 10.1016/j.scs.2020.102052.

[26] N. Al-Taleb and N. A. Saqib, “Towards a hybrid machine learning model for intelligent cyber threat identification in smart city environments,” Applied Sciences, vol. 12, no. 4, p. 1863, 2022. doi: 10.3390/app12041863 

[29] A. Kulshrestha, V. Krishnaswamy, and M. Sharma, “Bayesian BILSTM approach for tourism demand forecasting,” Ann Tour Res, vol. 83, p. 102925, 2020. doi: 10.1016/j.annals.2020.102925.

[39] L. Rice, E. Wong, and J. Z. Kolter, “Overfitting in adversarially robust deep learning,” 37th International Conference on Machine Learning, ICML 2020, vol. PartF16814, pp. 8049–8074, 2020.

[40] P. J. Brockwell and R. A. Davis, “Introduction to Time Series and Forecasting - Second Edition,” Springer-Verlag, p. 449, 2002. doi: 10.1007/978-3-319-29854-2

[47] R. Leardi, R. Boggia, and M. Terrile, “Genetic algorithms as a strategy for feature selection,” J Chemom, vol. 6, no. 5, pp. 267–281, 1992. doi: 10.1002/cem.1180060506 

We extend our heartfelt gratitude to the editor and reviewers for their thorough review and insightful comments. Your constructive feedback has been invaluable in enhancing the quality of our manuscript. Thank you for your time and effort in helping us improve our work.

---

## [Editor Report · Decision Letter 2]

2 Sep 2024

An Ensemble Deep Learning Framework for Energy Demand Forecasting Using Genetic Algorithm-Based Feature Selection

PONE-D-24-12727R2

Dear Dr. Siddiqui,

We’re pleased to inform you that your manuscript has been judged scientifically suitable for publication and will be formally accepted for publication once it meets all outstanding technical requirements.

Kind regards,

Sibarama Panigrahi, PhD

Academic Editor

PLOS ONE

Additional Editor Comments (optional):

The authors have satisfactorily addressed the suggestions made by the reviewers and improved the linguistic quality further. This resulted in overall quality improvement of the manuscript. I recommend accepting the manuscript in its present form.
---

## [Editor Report · Acceptance letter]

6 Sep 2024

PONE-D-24-12727R2 

PLOS ONE

Dear Dr. Siddiqui, 

I'm pleased to inform you that your manuscript has been deemed suitable for publication in PLOS ONE. Congratulations! Your manuscript is now being handed over to our production team.

Kind regards, 

on behalf of

Dr. Sibarama Panigrahi 

Academic Editor

PLOS ONE